# Molecular mechanisms of the CdnG-Cap5 antiphage defense system employing 3′,2′-cGAMP as the second messenger

Shirin Fatma[1,2], Arpita Chakravarti[1,2], Xuankun Zeng[1] & Raven H. Huang [1✉]

Cyclic-oligonucleotide-based antiphage signaling systems (CBASS) are diverse and abundant in bacteria. Here, we present the biochemical and structural characterization of two CBASS systems, composed of CdnG and Cap5, from *Asticcacaulis sp.* and *Lactococcus lactis*. We show that CdnG from *Asticcacaulis sp.* synthesizes 3′,2′-cGAMP in vitro, and 3′,2′-cGAMP is the biological signaling molecule that activates Cap5 for DNA degradation. Crystal structures of Cap5, together with the SAVED domain in complex with 3′,2′-cGAMP, provide insight into the architecture of Cap5 as well as molecular recognition of 3′,2′-cGAMP by the SAVED domain of Cap5. Amino acid conservation of the SAVED domain of Cap5, together with mutational studies, led us to propose a mechanism of Back-to-Front stacking of two SAVED domains, mediated by 3′,2′-cGAMP, to activate HNH nuclease domain for DNA degradation. This study of the most abundant CBASS system provides insights into the mechanisms employed by bacteria in their conflicts against phage.

[1] Department of Biochemistry, University of Illinois at Urbana-Champaign, Urbana, IL 61801, USA. [2] These authors contributed equally: Shirin Fatma and Arpita Chakravarti. ✉email: huang@illinois.edu

2′,3′-cGAMP, synthesized by cyclic GMP-AMP synthase (cGAS)[1], represents the first cyclic di-nucleotide to be involved in a self-defense system. As a nucleotidyl-transferase (NTase) of Polβ superfamily, cGAS is inactive when alone. Upon binding a double-stranded DNA (dsDNA) from viral or bacterial infection, cGAS becomes active and synthesizes 2′,3′-cGAMP second messenger using GTP and ATP as its substrates[1–3]. The signaling molecule cGAMP binds to STING (stimulator of interferon genes) receptor, triggering several STING-dependent signaling pathways that result in inhibition of viral or bacterial infection[4–7].

In bacteria, DncV was the first NTase to show synthesis of 3′,3′-cGAMP[8], which activates CapV to promote cell death[9]. Therefore, DncV-CapV represents the first characterized anti-phage defense system that shares a similar mechanism to the eukaryotic cGAS-STING system. In fact, a recent study of the bacterial STING-like proteins indicates that the eukaryotic cGAS-STING system might have its evolutionary origin from bacteria[10]. Recent studies by Sorek and colleagues indicate that the antiphage systems similar to DncV-CapV are abundant in bacteria, and these systems are collectively named cyclic-oligonucleotide-based antiphage signaling systems (CBASS)[11,12].

Despite diverse configurations of CBASS systems, two components are constant: enzymes that generate cyclic oligonucleotides, and effectors that are activated by cyclic oligonucleotides to exert their antiphage activities. Among various effectors of CBASS systems, the ones employing SAVED domain as the sensor of cyclic oligonucleotides constitute the biggest group[12], and within the SAVED domain-containing effectors, approximately 50% employ HNH nuclease domain for their antiphage activities[13]. Since the initial discovery of NTase-based antiphage systems via bioinformatic analysis[14], several CBASS systems have been recently characterized[10,11,13,15–17]. Perhaps the most relevant to our work described here is a recent study by Kranzusch and colleagues[13], as it represents the first characterization of SAVED domain-containing effectors.

In this work, we report biochemical and structural characterization of two Cap5 effectors composed of SAVED and HNH domains as well as their cognate CdnG proteins that synthesize the signaling molecule. We show that CdnG synthesizes 3′,2′-cGAMP, which specifically activates Cap5 for DNA degradation. Crystal structures of Cap5, as well as SAVED domain in complex with 3′,2′-cGAMP provide molecular insight into activation of Cap5 by 3′,2′-cGAMP.

## Results

### Bioinformatic analysis of effectors employing SAVED domains as sensors.

We have been interested in studying the effectors that contain SAVED domains. To provide guidance for the target selection of the study, we constructed a Sequence Similarity Network (SSN) of Pfam PF19145, which is the protein family of SAVED domain. SSN revealed many clusters of SAVED domain-containing effectors (Supplementary Fig. 1). Because Cap5 constitutes the biggest cluster, we, therefore, selected two Cap5 and their associated CdnG for our study (Fig. 1a, colored red and blue). In addition to genes encoding CdnG and Cap5, we found the operons might encode a third conserved protein of unknown function (Fig. 1a, colored green). Based on the recent classification of CBASS systems[12], the system under our study belongs to CBASS Type II (short). Because Sorek and colleagues employed short E2-like proteins (PF14457) for their classification of CBASS Type II (short)[12], the three-gene operons depicted in Fig. 1a appear to represent a new family of Type II (short) system because both proteins encoded by the third genes belong to a new protein family without a Pfam number. Functional

characterization of the third gene products is beyond the scope of this study. Instead, we focus our in vitro biochemical and structural studies on CdnG and Cap5.

### AsCdnG synthesizes 3′,2′-cGAMP in vitro.

With the exception of a few constitutively active CD-NTases such as DncV, most CD-NTases require activation, and the mechanisms of their in vivo activation are poorly understood[12]. Here we explore the potentially "leaky" activity of CdnG by employing different divalent ions as cofactors to investigate their possible products. Therefore, we cloned, expressed, and purified recombinant AsCdnG and LlCdnG, followed by performing in vitro reconstitution of their enzymatic activities. To simplify interpretation of the results, we added alkaline phosphatase to the reaction mixtures to remove 5′-terminal phosphate groups of all nucleotides before UPLC analysis (Fig. 1b, c). When $Mg^{2+}$ ion was employed as the cofactor, two compounds were synthesized by AsCdnG after 8 h of the reaction at 25 °C, but the yields were low (Fig. 1b, upper panel). The production of these two compounds was significantly improved by replacing $Mg^{2+}$ with $Mn^{2+}$ as the cofactor (Fig. 1b, lower panel). High-resolution LC-MS analysis identified these two compounds as cGAMP and ApG/GpA, respectively (Supplementary Fig. 2a, b). ApG/GpA is presumably the dephosphorylated product of pppApG/pppGpA, a potential reaction intermediate. We also carried out the same reactions with LlCdnG (Fig. 1c), which shares 30% sequence identity with AsCdnG. Compared to AsCdnG, LlCdnG is significantly less "leaky". Using $Mg^{2+}$ ion as the cofactor, only a tiny amount of ApG/GpA was observed (Fig. 1c, upper panel). The amount of ApG/GpA increased slightly when $Mn^{2+}$ ion was employed as the cofactor, which also allowed detection of a trace amount of cGAMP (Fig. 1c, lower panel).

To provide mechanistic insight into the enzymatic reaction carried out by AsCdnG, we performed further UPLC analysis (Supplementary Fig. 2c−e). We first compared cGAMP synthesized by AsCdnG to four commercially purchased cGAMP, revealing that AsCdnG-synthesized cGAMP is 3′,2′-cGAMP (Supplementary Fig. 2c). We next analyzed enzymatic digests of ApG/GpA by P1 and SVPD nucleases, respectively, identifying the 2′−5′ phosphate linkage in ApG/GpA (Supplementary Fig. 2d). Because the SVPD-digested products are GMP and A (Supplementary Fig. 2d, bottom panel), the identity of the second compound is ApG. Finally, we carried out the time course of the AsCdnG-catalyzed reaction (Supplementary Fig. 2e), which provided further support that pppApG is the reaction intermediate. Therefore, the sequential events of the AsCdnG-catalyzed reaction are shown in Fig. 1d. First, the 2′-OH of ATP carries out nucleophilic attack at α-phosphate of GTP, resulting in the formation of pppApG intermediate (Fig. 1d, Step 1). This is followed by nucleophilic attack of 3′-OH of the GMP moiety at α-phosphate of the ATP moiety, producing 3′,2′-cGAMP as the final product (Fig. 1d, Step 2).

### 3′,2′-cGAMP is the biological signaling molecule that activates AsCap5 and LlCap5 for DNA degradation.

To investigate whether 3′,2′-cGAMP is the second messenger of the CdnG-Cap5 antiphage defense system, we obtained recombinant AsCap5 and LlCap5 and performed in vitro DNA degradation assays (Fig. 2a, b). Linearized DNA plasmid (25 nM) was employed as the substrate, and 50 nM of Cap5 was used for the assays. In the absence of 3′,2′-cGAMP, both AsCap5 and LlCap5 are inactive (Fig. 2a, b, lane 1). Significant DNA degradation was observed in the presence of 3′,2′-cGAMP at a concentration as low as 10 nM (Fig. 2a, b, lane 3), and DNA degradation was complete when 100 nM of 3′,2′-cGAMP was employed (Fig. 2a, b, lane 4).

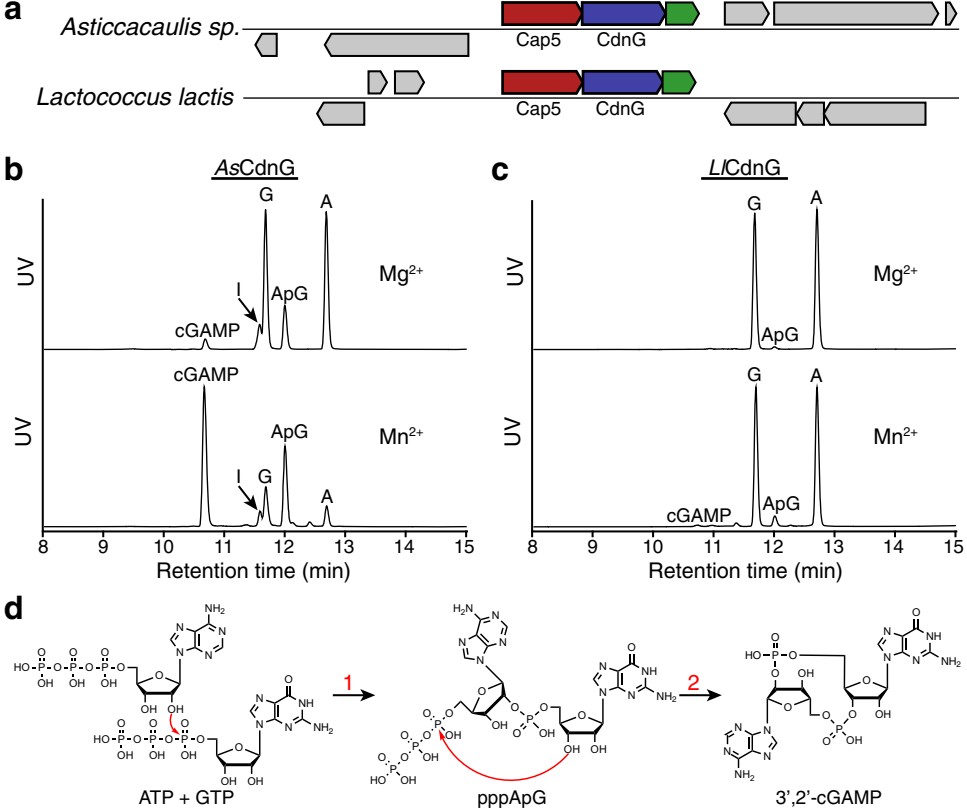

**Fig. 1 In vitro reconstitution of the enzymatic activity of CdnG. a** Schematic view of two operons encoding CdnG-Cap5 antiphage defense system. In addition to the genes encoding Cap5 and CdnG, a third gene encoding a conserved protein of unknown function might also be part of the operon (colored green). **b** UPLC analysis of the reaction mixtures carried out by *As*CdnG. ApG is presumably the dephosphorylated product of the reaction intermediate pppApG. I, Inosine, which is presumably the dephosphorylated product of ITP resulting from deamination of ATP. Results are representative of three independent experiments. **c** UPLC analysis of the reaction mixtures carried out by *Ll*CdnG. Results are representative of three independent experiments. **d** Proposed chemical steps of *As*CdnG-catalyzed reaction to produce 3',2'-cGAMP as the final product.

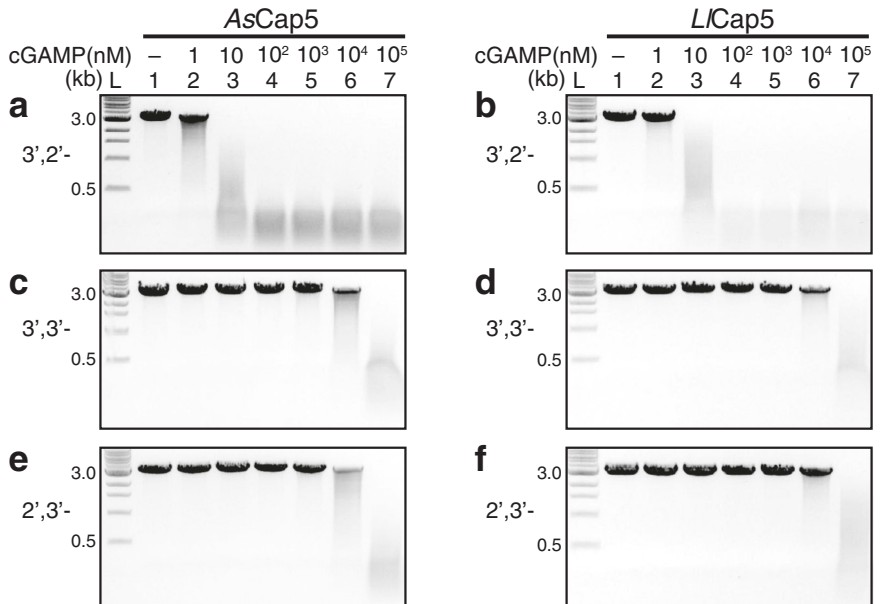

**Fig. 2 In vitro reconstitution of the enzymatic activity of Cap5.** DNA degradations were carried out by *As*Cap5 and *Ll*Cap5 in the presence of increasing concentrations of second messengers (10-fold increase with each step). Three different cGAMP were tested to activate Cap5 for DNA degradation. They are 3',2'-cGAMP (**a**, **b**), 3',3'-cGAMP (**c**, **d**), and 2',3'-cGAMP (**e**, **f**). L, DNA ladder marked with the molecular weight in kilo-base pair (kb). Results are representative of three independent experiments. Source data are provided as a Source Data file.

To assess ligand specificity of *As*Cap5 and *Ll*Cap5, we also carried out the same assays employing both 3′,3′-cGAMP and 2′,3′-cGAMP as potential activators (Fig. 2c–f). 3′,3′-cGAMP has been shown to be synthesized by bacterial enzymes[8]. On the other hand, 2′,3′-cGAMP is the product of cGAS[1], and has not yet been observed in bacteria. Compared to 3′,2′-cGAMP, both 3′,3′-cGAMP, and 2′,3′-cGAMP are significantly less effective. For example, it requires 100 μM of 3′,3′-cGAMP and 2′,3′-cGAMP to achieve the same degree of DNA degradation accomplished by 10 nM of 3′,2′-cGAMP (Fig. 2c–f, lane 7). Therefore, the ligand specificity of *As*Cap5 and *Ll*Cap5 for 3′,2′-cGAMP over 3′,3′-cGAMP and 2′,3′-cGAMP is approximately 10,000-fold. We also tested the possible activation of *As*Cap5 and *Ll*Cap5 by 3′,3′-c-di-GMP and 3′,3′-c-di-AMP, and both are even less effective than 3′,3′-cGAMP (Supplementary Fig. 3). The high selectivity of Cap5 for 3′,2′-cGAMP, together with the fact that it is synthesized by its cognate CdnG, strongly indicates that 3′,2′-cGAMP is the biological signaling molecule that activates *As*Cap5 and *Ll*Cap5 for DNA degradation in vivo.

**Structures of inactive *As*Cap5 and *Ll*Cap5.** To provide structural insight of Cap5, we solved the crystal structures of both *Ll*Cap5 and *As*Cap5 (Fig. 3a, b). The structures revealed that both Cap5 form homodimers, resulting from the side-by-side packing of two HNH domains. Each monomer of Cap5 is composed of the N-terminal HNH domain, the C-terminal SAVED domain, and a short flexible linker connecting these two domains. As described in later sections, the SAVED domain of Cap5 is likely to employ the interaction of two opposite sides to activate Cap5 for DNA degradation. Therefore, to facilitate structural description, we assigned the side where the ligand binds as Front, and the opposite side as Back (Fig. 3a, b).

The overall structures of *Ll*Cap5 and *As*Cap5 are very different (Fig. 3a, b). The different locations of two SAVED domains are responsible for the difference, as the overall structures of the HNH homodimers are similar (Supplementary Fig. 4b). In the structure of *Ll*Cap5 homodimer, two SAVED domains do not make any contact, and they are far away from each other (Fig. 3a). Therefore, the structure of *Ll*Cap5 can be described as in open configuration. On the other hand, two SAVED domains in the structure of *As*Cap5 are close and make contact with one another. But the two contacting surfaces are not completely closed and a gap between them can be seen (Fig. 3b). Therefore, the structure of *As*Cap5 can be best described as in ajar configuration. Importantly, unlike the two-fold symmetry applicable for the entire *Ll*Cap5 structure, the Front of the first SAVED domain of *As*Cap5 faces the Back of the second SAVED domain (Fig. 3b). Therefore, the two-fold symmetry is only applicable to the HNH homodimer.

The detailed structural analysis of the SAVED domains will be carried out below. Here, we focus our analysis on the structure of the HNH domains. Although the overall HNH homodimers of these two structures are similar (Supplementary Fig. 4b), there are significant differences locally. The HNH domain of *As*Cap5 (*As*Cap5-HNH) is significantly more structured near the active site than the one of *Ll*Cap5 (Supplementary Fig. 4c, d). Specifically, a well-coordinated Mg²⁺ ion is only observed with *As*Cap5-HNH (Supplementary Fig. 4c), and nine residues of *Ll*Cap5-HNH near the active site are disordered (Supplementary Fig. 4d). Therefore, it is reasonable to conclude that the configuration of *Ll*Cap5 shown in Fig. 3a is enzymatically inactive.

To provide functional insight into the HNH domain of *As*Cap5, we performed Dali structural search[18] based on the

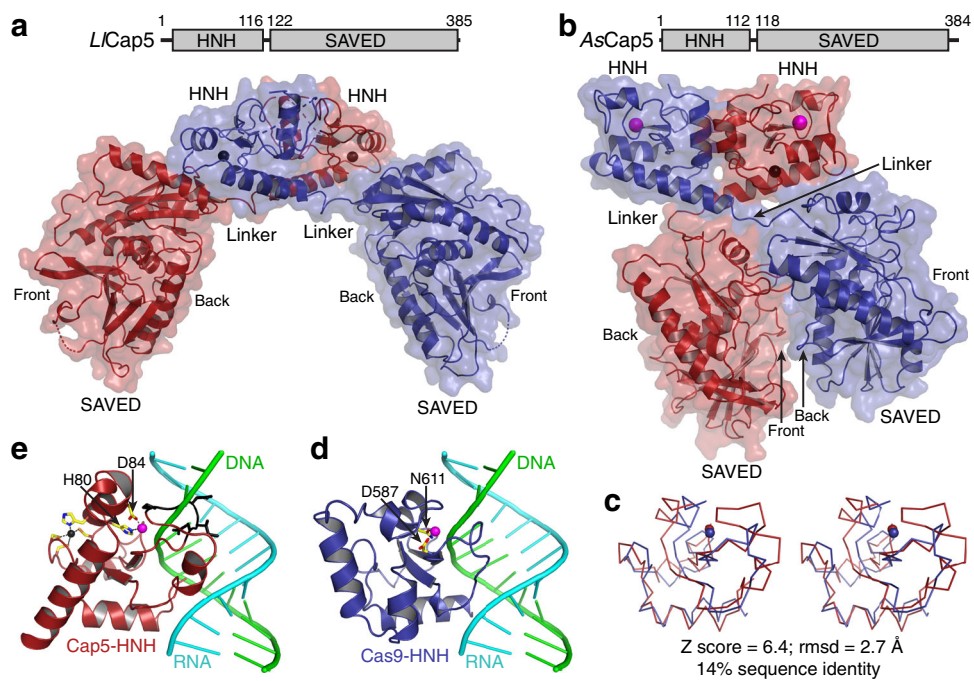

**Fig. 3 Crystal structures of unliganded Cap5. a** Cartoon and surface depiction of the structure of *Ll*Cap5. Cap5 forms a homodimer, and two monomers are colored red and blue, respectively. Zn²⁺ ions are in the sphere and colored black. **b** Structure of *As*Cap5. Additional Mg²⁺ ions in the active sites of HNH domains are in sphere and colored magenta. **c** Stereoview of structural superposition of HNH domain from *As*Cap5 (red) to the one from Cas9 (blue). The structures are depicted in ribbon, and Mg²⁺ ions are in sphere. **d** Structure of Cas9-HNH domain in complex with DNA●RNA hybrid. Two residues coordinating a Mg²⁺ ion are in stick and colored individually, with carbon in yellow, oxygen in red, and nitrogen in blue. **e** Structure of *As*Cap5-HNH domain with the modeled DNA●RNA hybrid from **d**. In addition to coordination of a Mg²⁺ ion, a zinc finger, which is absent in Cas9-HNH domain, is also depicted in the stick, with the Zn²⁺ ion in black and the sulfur atom in orange. The peptide having a steric clash with the DNA strand of the modeled DNA●RNA hybrid is colored black and depicted in the stick.

structure of *As*Cap5-HNH. Among top 30 hits, approximately half are the structures of the HNH domain of Cas9 (Cas9-HNH). Among those 30 structures, only one (PDB accession code 6JDV) shows the presence of nucleic acids in the active site of the HNH domain[19]. Therefore, we employed this Cas9-HNH for our structural analysis of *As*Cap5-HNH. Structural superposition indicates the front portion of the HNH domains align well, including the coordinated $Mg^{2+}$ ions (Fig. 3c). They differ on the back portion, with a helix-turn-helix motif for Cas9-HNH (Fig. 3d) and a long helix for *As*Cap5-HNH (Fig. 3e). The structural divergence on the back probably reflects different mechanisms of HNH activation by Cap5 and Cas9. We created a model of *As*Cap5-HNH in complex with DNA•RNA hybrid, which revealed that part of a loop in *As*Cap5-HNH has a steric clash with the DNA strand of the modeled DNA•RNA hybrid (Fig. 3e, colored black). Based on this analysis, we conclude that the configuration of *As*Cap5 shown in Fig. 3b is also likely to be inactive. We acknowledge that the modeled DNA•RNA hybrid is structurally different from the double-stranded DNA, which is the substrate of Cap5. However, the structural difference of A and B-forms DNAs is small, and a similar steric clash would likely be observed with a modeled double-stranded DNA.

**Molecular recognition of 3′,2′-cGAMP by the SAVED domain of Cap5.** To provide molecular insight into the ligand specificity of the SAVED domain of Cap5, we solved the crystal structure of the SAVED domain of *Ll*Cap5 (*Ll*Cap5-SAVED) in complex with 3′,2′-cGAMP (Fig. 4a and Supplementary Figs. 5, 6). The structures of the SAVED domains of *Ll*Cap5-SAVED and the full-length *Ll*Cap5 are essentially the same (rmsd = 0.4 Å). In addition, despite of a modest sequence identity (23%) between the SAVED domains of *Ll*Cap5 and *As*Cap5, the structure of *Ll*Cap5-SAVED is highly homologous to the structure of the SAVED domain of *As*Cap5 (rmsd = 1.9 Å, Supplementary Fig. 5b).

Kranzusch and colleagues solved the first structure of SAVED domain, and it was depicted as a fusion of two CARF-like motifs[13]. The evolutionary connection between SAVED and CARF domains was further confirmed by a recent comprehensive bioinformatic analysis of proteins containing these two domains[20]. In our structure, 3′,2′-cGAMP is found at the center of *Ll*Cap5-SAVED (Fig. 4a). Employing a similar evolutionary approach to describe the structure of *Ll*Cap5-SAVED, 3′,2′-cGAMP is bound at the interface of these two CARF-like motifs (Supplementary Fig. 6). The critical R234 (see below) is in the loop that connects these two CARF-like motifs.

The high-resolution structure of the complex unambiguously confirms the chemical structure of cGAMP synthesized by *As*CdnG as 3′,2′-cGAMP (Fig. 4b). Several conserved residues of *Ll*Cap5-SAVED are involved in the recognition of 3′,2′-cGAMP (Fig. 4c). Specifically, the adenine base is stacked between the side chains of I208 and R234, which is further stacked by the side chain of F304. The side chain of R234 also forms a hydrogen bond with the 2′−5′ phosphate linkage. The involvement of the side chain of R234 for both base-stacking and phosphate-hydrogen bonding appears to be only feasible with the 2′−5′ phosphate linkage, indicating R234 is the discriminating residue to distinguish 3′,2′-cGAMP from 3′,3′-cGAMP. In addition to R234, the side chain of S274 also hydrogen bonds with the 2′−5′ phosphate linkage, and the free 3′-OH group of the AMP moiety hydrogen bonds with the main-chain carbonyl group of T344 (Fig. 4c). On the other hand, a single residue, R281, is responsible for the recognition of the guanine base (Fig. 4c). Interestingly, both N6 and the 3′−5′ phosphate linkage do not make any specific contacts with the residues of the SAVED domain, presenting the possibility that these two functional groups might

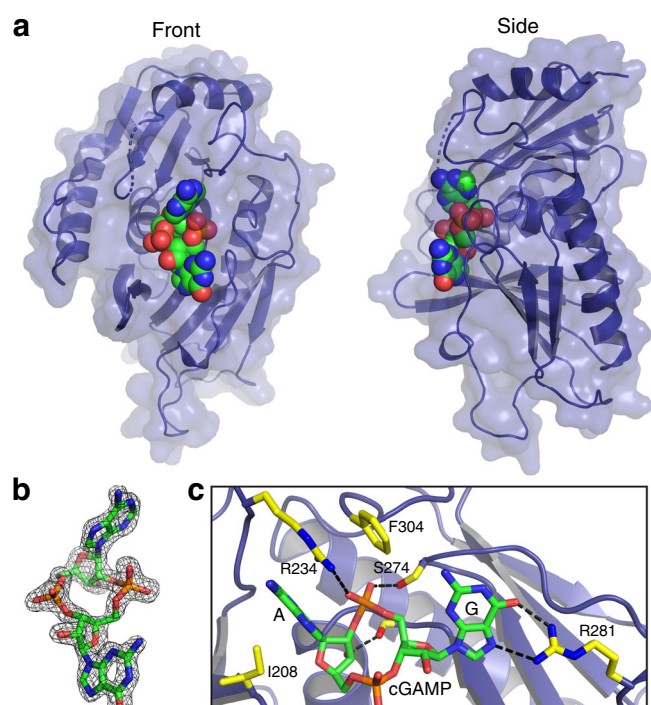

**Fig. 4 Molecular recognition of 3′,2′-cGAMP by Cap5-SAVED. a** Crystal structure of *Ll*Cap5-SAVED in complex with 3′,2′-cGAMP. *Ll*Cap5-SAVED is in cartoon and surface, and 3′,2′-cGAMP is in the sphere and colored individually, with carbon in green, oxygen in red, nitrogen in blue, and phosphate in orange. **b** Simulated-annealing $F_o−F_c$ omit map (contoured at 2σ) of the ligand density, demonstrating unambiguous assignment of the second messenger as 3′,2′-cGAMP. **c** Detailed interactions between 3′,2′-cGAMP and residues from *Ll*Cap5-SAVED. The side chains of *Ll*Cap5-SAVED are in stick and colored individually, with carbon in yellow, oxygen in red, and nitrogen in blue. Hydrogen bonds between 3′,2′-cGAMP and *Ll*Cap5-SAVED are highlighted in black dashed lines.

be involved in interacting with the second SAVED domain to activate Cap5, as discussed below.

**Proposed mechanism of 3′,2′-cGAMP activating Cap5 for DNA degradation.** Despite our extensive effort, we were not able to obtain a crystal structure of the full-length Cap5 in complex with 3′,2′-cGAMP. Here, we employ a combination of amino acid conservation of Cap5 and mutagenesis of *As*Cap5 to probe the possible mechanism of 3′,2′-cGAMP activating Cap5 for DNA degradation. We first carried out structural analysis of *As*Cap5 via ConSurf[21]. Specifically, the amino acid sequence of *As*Cap5 was first aligned with the sequences of 200 additional Cap5 that share 35−95% sequence identities with *As*Cap5. This was followed by coloring each residue of *As*Cap5 according to the degree of conservation (Fig. 5a−e). This analysis revealed that two additional sides of SAVED domains other than the Front and Back are not conserved (Fig. 5a). On the other hand, the central regions of both the Front and Back are highly conserved (Fig. 5b, c), indicating that these two regions are functionally important. The rationale for the conservation on the Front is obvious, as it is where 3′,2′-cGAMP binds (Fig. 5d). On the other hand, the purpose of conservation on the Back is unclear. Because these two conserved regions face each other based on the structure of *As*Cap5 (Fig. 5a−c), we hypothesize that these two regions interact with each other, mediated by 3′,2′-cGAMP, and the structural change resulted from the interaction activates Cap5 for DNA degradation.

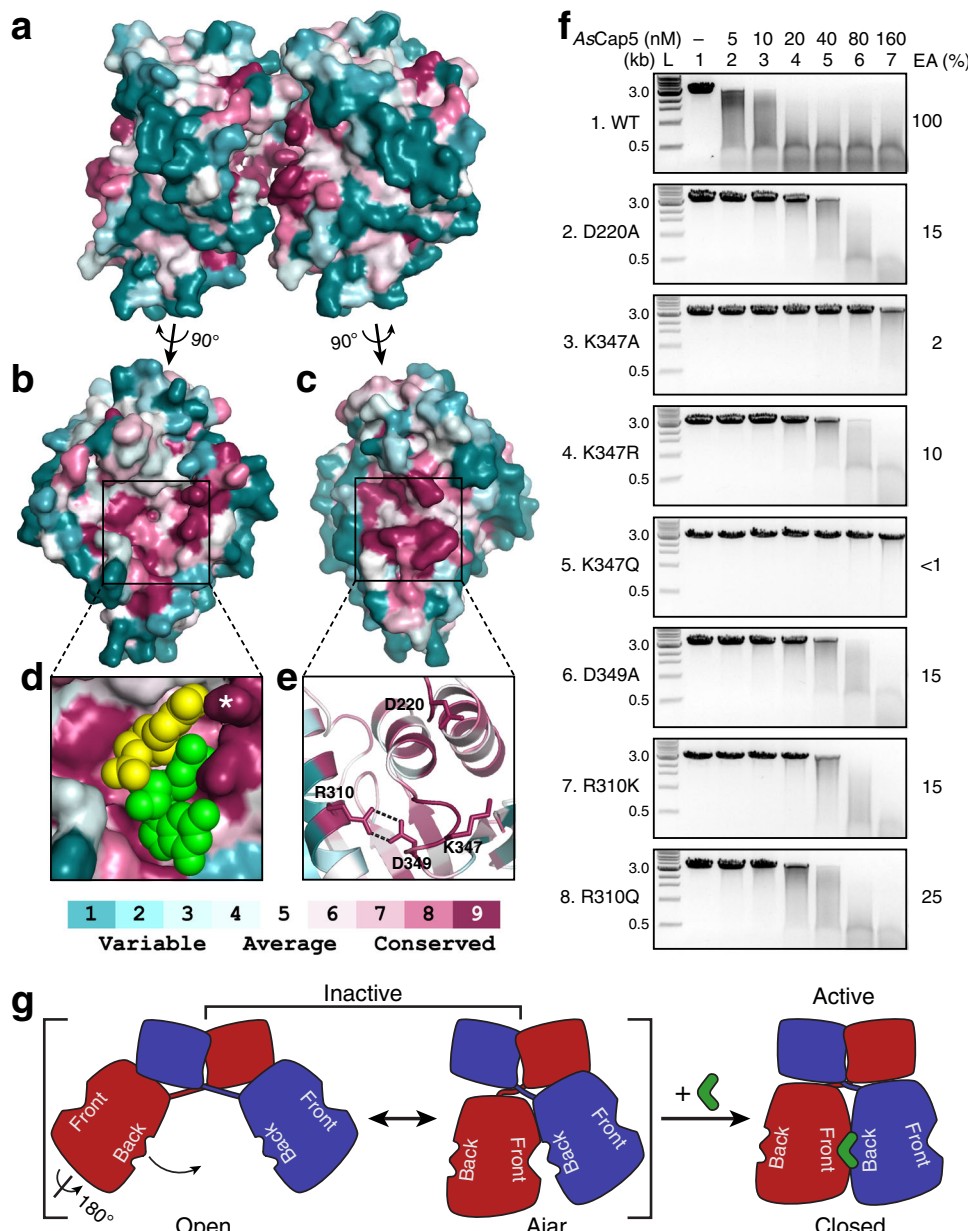

**Fig. 5 Amino acid conservation of Cap5-SAVED and mutational studies. a** Surface depiction of the structure of *As*Cap5-SAVED homodimer viewed from side. Two SAVED domains are colored based on the degree of amino acid conservation. **b, c** Conservation on the Front and Back of *As*Cap5-SAVED, respectively. **d** Closeup view of the ligand-binding pocket on the Front together with the docked 3′,2′-cGAMP. 3′,2′-cGAMP is in the sphere, with AMP and GMP moieties colored yellow and green, respectively. The side chain of R231 is marked with a white asterisk. **e** Cartoon depiction of the closeup view of the conserved residues on the Back. **f** DNA degradation assays by *As*Cap5 mutants. L, DNA ladder marked with the molecular weight in kilo-base pair; EA, estimated activity. Results are representative of three independent experiments. Source data are provided as a Source Data file. **g** Proposed mechanism of Cap5 activation via cGAMP-induced Back-to-Front stacking of two SAVED domains.

To test this hypothesis, we selected four highly conserved residues for mutational studies (Fig. 5e). D220 and K347 are located at the tip of two protruding loops, which might make direct contact to 3′,2′-cGAMP (Supplementary Fig. 7). The side chains of R310 and D349 form two hydrogen bonds (Fig. 5e), and their role might assist potential interaction between K347 and 3′,2′-cGAMP. D220A mutation retains ~15% activity of *As*Cap5 (Fig. 5f, panel 2). On the other hand, the effect of K347A mutation is significantly more pronounced, resulting in ~98% reduction of activity (Fig. 5f, panel 3). A conservative K347R mutation recovered some of the lost activity of the K347A mutant, but it is still approximately 10-fold less active compared to the wild-type enzyme (Fig. 5f, panel 4). A second conservative

K347Q mutation unexpectedly resulted in significantly less soluble protein. We managed to obtain a small amount of recombinant K347Q mutant, and DNA degradation assays showed that it was even less active than K347A mutant (Fig. 5f, compare panel 5 to panel 3). D349A mutation showed a similar degree of reduced activity as D220A mutation (Fig. 5f, panel 6). On the other hand, the R310A mutant was completely insoluble. Nevertheless, we were able to obtain recombinant proteins of more conservative R310K and R310Q mutants, and both also showed significantly reduced activities (Fig. 5f, panels 7 and 8). In summary, mutation of each of these four residues has a detrimental effect on *As*Cap5 activity, with the most negative effect from K347 mutants.

Based on the collective biochemical and structural data we have presented, we propose a mechanism of Cap5 activation for DNA degradation as schematically depicted in Fig. 5g. In the absence of 3′,2′-cGAMP, Cap5 homodimer is inactive and structurally heterogenous in solution. This is mainly caused by two mobile SAVED domains. Two structures of Cap5 shown in Fig. 3a, b probably capture the two ends of the spectrum of Cap5 conformers: an Open configuration represented by the structure of *Ll*Cap5, and an Ajar configuration represented by the structure of *As*Cap5 (Fig. 5g, left). 3′,2′-cGAMP might act as a glue, bringing two SAVED domains closer to seal the gap shown in the structure of *As*Cap5 (Fig. 5g, right). The movement of the SAVED domains somehow propagates the reorganization of the structure near the active site of one or both HNH domains, activating HNH domain(s) for DNA degradation. We recognize that such a Back-to-Front stacking of SAVED domains has the potential for SAVED domain-containing effectors to form oligomers. This is probably what happened with Cap4[13]. In our case, however, the Back-to-Front stacking of SAVED domains appears only for dimerization, consistent with the chromatographic analysis of Cap5 in the presence of 3′,2′-cGAMP (Supplementary Fig. 8).

## Discussion

In this study, we have reported the biochemical characterization of a member of the G clade of CD-NTases, and evidence for the enzymatic synthesis of 3′,2′-cGAMP in bacteria. We have demonstrated that *As*CdnG synthesizes 3′,2′-cGAMP in vitro, and 3′,2′-cGAMP is the biological signaling molecule that activates both *As*Cap5 and *Ll*Cap5 for DNA degradation. While this manuscript was in review, two publications revealed enzymatic synthesis of 3′,2′-cGAMP by cGAS-like receptors from *Drosophila*[22,23]. In addition, we have also carried out structural and mutational studies of Cap5, which provide structural insight into Cap5, as well as the mode of action of the SAVED domain activating Cap5 for DNA degradation. Here, we discuss our findings in a broader context of proteins of the SAVED family in general, including the comparison of SAVED domains from different subfamilies, ligand specificities of the SAVED domains of Cap5, and comparing the mode of action of SAVED domain to other sensors that also employ cyclic oligonucleotides as signaling molecules.

We first compare the structure of the SAVED domain of Cap5 to the one of Cap4. Dali structural search based on *Ll*Cap5-SAVED revealed the top scores are the structures of the SAVED domains of Cap4 (Cap4-SAVED), recently reported by Kranzusch and colleagues[13]. Because *Ab*Cap4-SAVED (PDB ID 6VM6) is in complex with 2′,3′,3′-cAAA, it was selected for our structural comparison. *Ll*Cap5-SAVED and *Ab*Cap4-SAVED only share 13% sequence identity, and rmsd of these two structures is 3.2 Å. Therefore, the SAVED domains of Cap5 and Cap4 are significantly different. This is consistent with our observation that *Ll*Cap5 and *Ab*Cap4 belong to different clusters of SSN (Supplementary Fig. 1). This is also consistent with a recent bioinformatic analysis by Makarova et al., classifying Cap5 and Cap4 to SAVED-1 and SAVED-3 subfamilies, respectively[20]. Structural superposition of *Ll*Cap5-SAVED and *Ab*Cap4-SAVED revealed that, although the conformations of bases differ, the central rings composed of ribose and phosphate linkages of both ligands are at approximately the same location (Supplementary Fig. 9). However, the details of how these two ligands are recognized by their respective SAVED domains differ. In particular, the 2′−5′ phosphate linkages of these two ligands are located at the opposite sides of the ring structures (Supplementary Fig. 9, highlighted in spheres), indicating

possibly different mechanisms of discriminating the ligand possessing 2′-5′ phosphate linkage from the one of 3′-5′ phosphate linkage. Furthermore, a longer loop between β7 and α6 in *Ll*Cap5-SAVED (Supplementary Fig. 5) presents the steric clash with the third nucleotide of 2′,3′,3′-cAAA (Supplementary Fig. 9, colored black), suggesting that this loop might be responsible for discriminating cyclic di-nucleotides from cyclic tri-nucleotides.

In addition to comprehensive characterization of Cap4, Kranzusch and colleagues also carried out in vitro (one Cap5) and in vivo (nine Cap5) functional assays of Cap5 in response to 3′,3′-cGAMP[13]. Unlike two Cap5 reported here, Cap5 from *Burkholderia pseudomallei* was able to degrade DNA in vitro in the presence of 3′,3′-cGAMP. Furthermore, among nine Cap5 tested in vivo, five of them were shown to be active in response to 3′,3′-cGAMP generated in vivo by the co-expressed DncV. Therefore, it is likely that different Cap5 might employ either 3′,2′-cGAMP or 3′,3′-cGAMP as their biological signaling molecules for their antiphage activities. We also cannot rule out the possibility that cyclic oligonucleotides other than 3′,2′-cGAMP and 3′,3′-cGAMP might also be the signaling molecules of Cap5. Given that Cap5 is the most abundant effectors of CBASS systems, it is worth carrying out the additional characterization of Cap5 similar to the ones reported here to provide a clearer picture of the ligand specificity of Cap5.

To date, cGAS[1], bacterial CD-NTases[8,10,13,15,16], and CRISPR-associated cyclases[24,25] have been shown to synthesize cyclic oligonucleotides. Structures of effectors in complex with some of these signaling molecules are available. They include eukaryotic and bacterial STING[10,26], bacterial endonuclease[17], and CRISPR-associated RNA and DNA nucleases[27−31]. All of them employ one surface where the ligand-binding pocket locates to recognize the signaling molecules. Therefore, the ligand-induced Back-to-Front stacking of SAVED domains appears to be unique. For Cap5 homodimer, only 50% of the capacity of this Back-to-Front stacking has been utilized (Fig. 5g). Therefore, the ligand-induced oligomerization, in particular the one that form a circle, would be more efficiently fulfilling the potential of this Back-to-Front stacking. In addition to nucleases, such as HNH in Cap5 and REase in Cap4, as effector domains, a significant number of SAVED domain-containing effectors employ two transmembrane helices (2TM)[20], which is predicted to form pore on the membrane of bacteria to promote cell death. Therefore, we predict that the pore formation of 2TM effectors is achieved by circular oligomerization of effectors via the ligand-induced Back-to-Front stacking of SAVED domains proposed here. In our view, the mechanism of ligand-induced Back-to-Front stacking of SAVED domains is likely to be universal for activating all SAVED domain-containing effectors, but significantly more studies are required to validate this hypothesis.

## Methods

**General materials and methods**. Synthetic cyclic dinucleotides used for biochemical assays: 3′,3′-cGAMP, 3′,3′-c-di-GMP, and 3′,3′c-di-AMP were purchased from Sigma-Aldrich, 2′,3′-cGAMP, 3′,2′-cGAMP, and 2′,2′-cGAMP were purchased from Axxora. All synthetic cyclic dinucleotides were further HPLC purified. Calf Intestinal Phosphatase (CIP), P1 nuclease, restriction digestion enzymes, and BSA were purchased from New England Biolabs (NEB). Snake venom phosphodiesterase (SVPD) was purchased from Sigma-Aldrich. CAPSO (Stock Options pH Buffer Kit) and 48-well hanging drop trays were purchased from Hampton Research. Pfu turbo DNA polymerase was purchased from Agilent. All the codon-optimized genes and oligonucleotides were purchased from Integrated DNA Technologies (IDT).

**Generation of sequence similarity network of SAVED domain-containing proteins**. Bioinformatic analyses were performed on the database of UniProt 2020_04 and InterPro 81. Calculations were carried out at the EFI website (https://efi.igb.illinois.edu/)[32]. PF18145 (Pfam of SAVED) was submitted for the initial

calculation for the Sequence Similarity Network (SSN) of the SAVED domain-containing proteins. SSN file was processed and displayed with Cytoscape, and yFiles Organic Layout was used for the layout of nodes and edges. Minor adjustments of the positions of a few clusters and single nodes were made to make nodes fit better within the space of the figure.

**Cloning, overexpression, and purification of recombinant proteins.** CdnG and Cap5 genes from *Asticcacaulis sp.* (*As*) and *Lactococcus lactis* (*Ll*) (codon-optimized for *Escherichia.coli*) were ordered from IDT. For structural studies of SAVED domain of *Ll*Cap5, Δ123*Ll*Cap5 construct was created by deleting the first 123 amino acids from the N-terminus. The synthetic genes were cloned into pRSF-1 vector, which carries an N-terminal 6XHis tag followed by a SUMO tag. *E. coli* BL21 (DE3) transformed with expression plasmids were grown at 37 °C until A$_{600}$ reached 0.4–0.6. After cooling to 18 °C, expression was induced with 0.5 mM isopropyl-β-D-thiogalactopyranoside (IPTG), and cells were grown at 18 °C overnight. The cultures were harvested and the cell pellets were resuspended in lysis buffer (20 mM Tris-HCl, pH 8.0, 500 mM NaCl, 5% glycerol). Cells were lysed using French Press, followed by centrifugation at 14,000 rpm for 50 min at 4 °C to remove cell debris. After filtration with 0.45 μm filter, the supernatant was loaded into HisTrap-FF column (GE Healthcare). The proteins were eluted using elution buffer (20 mM Tris-HCl, pH 8.0, 500 mM NaCl, 500 mM imidazole, 5% glycerol) and fractions containing SUMO tagged proteins were combined and cleaved with Ulp1 protease for 1 h. The salt concentration of the digested protein was increased to 1.9 M (NH$_4$)$_2$SO$_4$ and loaded onto a HiTrap-Phenyl-HP column (GE Healthcare) with buffer A [20 mM Tris-HCl pH 8.0, 1.25 M (NH$_4$)$_2$SO$_4$] and buffer B (20 mM Tris-HCl pH 8.0). The eluted proteins were then concentrated and further purified on a Superdex 200 size exclusion chromatography (GE Healthcare) column with storage buffer (20 mM HEPES pH 7.0, 200 mM NaCl, 2% glycerol). All *As*Cap5 mutants were purified using the above-mentioned protocol. Protein purity was assessed by SDS-PAGE with Coomassie staining and samples were flash-frozen in liquid nitrogen and stored at −80 °C.

**Size exclusion chromatography of Cap5 with 3′,2′-cGAMP.** Purified *As*Cap5 and *Ll*Cap5 were diluted in 20 mM HEPES, pH 7.0, 200 mM NaCl, 2% glycerol to a final concentration of 60 μM and 80 μM respectively. Samples were incubated on ice without or with 3′,2′-cGAMP (4X the concentration of protein) for 30 min followed by brief centrifugation (12,303 × g, 1 min, 4 °C) to remove precipitated protein before injection into Superdex 200 size exclusion column (GE Healthcare).

**In vitro reconstitution of enzymatic activity of CdnG.** For in vitro reconstitution assays, 150 μl reactions with 20 μM of recombinant enzyme and 200 μM each of ATP and GTP in 1X CdnG-Reaction buffer (50 mM CAPSO, pH 7.4, 50 mM KCl, 5 mM MgCl$_2$ or MnCl$_2$, 1 mM DTT) were carried out. Following 8 hours of incubation at room temperature, reactions were terminated using 10 U of Calf Intestinal Phosphatase and incubated at 37 °C for 1 h. The reaction mixture was then filtered using 10 kDa cut-off filters to remove the protein and the flow-through was analyzed using UPLC and LC-MS. For the time course, the reaction conditions were the same as described above except the reaction volume was increased to 1.5 ml and the reaction was stopped at various time points by adding 10 U of Calf Intestinal Phosphatase, followed by UPLC analysis and quantification of peaks corresponding to 3′,2′-cGAMP, the reaction intermediate ApG, A, G and I.

To identify the CdnG product, commercially available 2′,3′-cGAMP, 2′,2′-cGAMP, 3′,3′-cGAMP, and 3′,2′-cGAMP were made to solutions of 25 μM with 1X CdnG-Reaction buffer and injected into UPLC as control. This was followed by co-injection of 25 μM CdnG product with these synthetic controls. To identify the phosphate linkage of the CdnG product, purified reaction intermediate ApG was diluted to 100 μM in 150 μl reactions and supplemented with 1X P1 buffer (30 mM NaOAc, pH 5.3, 5 mM ZnSO$_4$, 50 mM NaCl) or 1X SVPD buffer (50 mM Tris pH 9.0, 10 mM MgCl$_2$, 50 mM NaCl). SVPD (5 mU) or P1 (100 mU) was added followed by incubation at 37 °C for 30 min or 1 h respectively. Reactions were then filtered using 10 kDa cut-off filters and flow-through was analyzed by UPLC.

UPLC analyses were carried out on a Waters Acquity Arc system with a reversed phase Luna Omega C18 column (150 × 2.1 mm; Phenomenex) with a flow rate of 0.2 ml/min. The column was equilibrated with 100% solvent A (5 mM ammonium acetate, pH 5.0), and the following gradient was applied with solvent B (100% acetonitrile): 0 min, 0% B; 1 min, 0% B; 15 min, 30% B; 18 min, 40% B; 20 min, 0% B.

**Purification of the product and reaction intermediate of the *As*CdnG-catalyzed reaction.** CdnG catalyzed reactions were scaled up using 50 μM enzyme and 1 mM each of ATP and GTP in 1X CdnG-Reaction buffer. Following overnight incubation at room temperature, the reaction was stopped using 50 U Calf Intestinal Phosphatase at 37 °C for 1 h. After filtration using 10 kDa cut-off filters, the flow-through was used to purify 3′,2′-cGAMP and ApG. Large scale CdnG product and intermediate were purified using a Waters 1525 HPLC system with a reversed phase Luna C18(2) column (250 × 2 mm; Phenomenex) with a flow rate of 0.3 ml/min. Mobile phase equilibration with 100% solvent A (5 mM ammonium

acetate, pH 5) was followed by the following gradient with solvent B (40% acetonitrile): 0 min, 0% B; 0.5 min, 0% B; 3 min, 15% B; 20 min, 20% B; 35 min, 60% B; 40 min, 60% B; 45 min, 0% B; 50 min, 0% B. Samples were dried by speed-vac concentrator for approximately 4 h and resuspended in nuclease-free water. Concentrations were measured by UV-visible spectroscopy on a Nanodrop 2000 spectrophotometer. All synthetic cyclic dinucleotides were also purified using the above-mentioned protocol.

**DNA degradation assays by Cap5.** For all DNA degradation assays, EcoRI-linearized ZIKA minigenome pUC19 plasmid (3,276 bp) was used. Assays were performed by incubating 50 nM *As*Cap5 and *Ll*Cap5 with different concentrations of cyclic dinucleotides (3′,2′-cGAMP, 3′,3′-cGAMP, 2′,3′-cGAMP, 3′,3′-c-di-GMP, and 3′,3′-c-di-AMP) on ice for 10 min in a Cap5-Reaction buffer (50 mM Tris-HCl, pH 7.4, 1 mM MgCl$_2$, 0.5 mM MnCl$_2$, and 50 μg/mL BSA) in a final reaction volume of 10 μl. The degradation reaction was initiated by the addition of 25 nM plasmid substrate, followed by incubation at 37 °C for 15 min. Reactions were stopped by addition of 6X Loading buffer (15% Ficoll-400, 60 mM EDTA, 19.8 mM Tris-HCl pH 8.0, 0.48% SDS, 0.12% Dye1, and 0.006% Dye2), and then 10 μL was separated on a 1% TBE (Tris-Borate-EDTA) agarose gel. Gels were run at 120 V for 20 min, then stained with ethidium bromide and imaged by UV illumination. For mutational assays, same protocol was followed with different concentrations of purified mutants and 10 nM of 3′,2′-cGAMP. Data are representative of three independent experiments.

**Site-directed mutagenesis of *As*Cap5.** Individual point mutations were introduced into *As*Cap5 plasmid by Quick-change mutagenesis according to the direction provided by Agilent. Oligonucleotides employed for creating mutants were ordered from IDT (Supplementary Table 2). All mutations were verified by DNA sequencing of the entire gene prior to use.

**Crystallization, data collection, and structural determination.** *As*Cap5 and *Ll*Cap5 in *apo* form and Δ123*Ll*Cap5 in complex with 3′,2′-cGAMP were crystallized at 18 °C using the hanging drop vapor diffusion method. Concentrated protein stocks were diluted in Dilution buffer (20 mM HEPES, pH 7.0, 200 mM NaCl, 2% glycerol) to final concentrations. In all cases, optimized crystals were obtained using 48-well hanging drop tray in 2 μl drops mixed 1:1 over a 200 ul reservoir solution. Final optimized crystal growth conditions were as follows: (1) Crystals of native or selenomethionine substituted *apoAs*Cap5 grew at 6 mg/ml in 15–17% PEG6000, 100 mM Tris-HCl, pH 8.5, 100 mM NaCl; 2) Crystals of native *apoLl*Cap5 grew at 5 mg/ml in 12–15% PEG6000, 100 mM sodium citrate, pH 5.5, 100 mM NaCl; (3) Native Δ123*Ll*Cap5 (5 mg/ml) was pre-incubated with 0.8 mM of 3′,2′-cGAMP and crystals grew in 10–12.5% PEG6000, 100 mM Tris-HCl, pH 8.5. Crystals appeared after 2−40 days and were cryoprotected using reservoir solution supplemented with 25% glycerol.

Native and single-wavelength anomalous dispersion (SAD) data were collected at the 21-ID beamlines at the Advanced Proton Source (APS). Data were processed using the HKL2000 program[33]. To solve the structure of *As*Cap5, the phase was determined based on selenomethionine single-wavelength anomalous diffraction data using the Phenix program[34]. A partial model was automatically built using Phenix. The remaining model was manually built using the Coot program[35]. Refinement was done using Phenix. The initial phasing of the structure of *Ll*Cap5-SAVED in complex with 3′,2′-cGAMP was obtained using the Molecular Replacement method in Phenix, using the structure of the SAVED domain of *As*Cap5 as the search model. The initial phase of *Ll*Cap5 was also determined using the Molecular Replacement method in Phenix, using the structure of *Ll*Cap5-SAVED as the search model. Model building and refinement of the structure of *Ll*Cap5-SAVED–3′,2′-cGAMP and *Ll*Cap5 were carried out similarly for the structure of *As*Cap5. Figures were prepared using PyMOL, GraphicConverter version 11, and Adobe Illustrator 2021. Representative electron density maps are provided in the Source Data file.

**Reporting summary.** Further information on research design is available in the Nature Research Reporting Summary linked to this article.

## Data availability
Coordinates and structural factors have been deposited in the Protein Data Bank (PDB) under accession codes of 7RWK for *As*Cap5, 7RWM for *Ll*Cap5, and 7RWS for *Ll*Cap5-SAVED in complex with 3′,2′-cGAMP. The coordinates of the structure of Cas9 in complex with a DNA•RNA hybrid, employed for our comparison to the structure of *As*Cap5, are available in the PDB under accession code 6JDV. The previously published structure of *Ab*Cap4-SAVED in complex with 2′,3′,3′-cAAA is available in the PDB under accession code 6VM6. All relevant data are available from the authors. Source data are provided with this paper.

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

## Acknowledgements

This work was supported by the National Institutes of Health (grant AI150229 and GM120764 to R.H.H.). We thank F. Sun for assistance with LC-MS, Z. Wawrzak, and S. Anderson for help with data collection.

## Author contributions

R.H.H. conceived of the project. A.C. carried out in vitro reconstitution of CdnG. S.F. performed DNA degradation assays by Cap5. S.F. and A.C. carried out crystallization of Cap5 and data collection. R.H.H. solved crystal structures of Cap5. X.Z. carried out enzymatic assays of CD-NTases related to CdnG. R.H.H. wrote the manuscript with the input from all authors.

## Competing interests

The authors declare no competing interests.
