## [Peer Review File · Nature Communications]

REVIEWER COMMENTS

Reviewer #1 (Remarks to the Author):

Molecular mechanisms of the CdnG-Cap5 antiphage defense system employing 2 3',2'-cGAMP as the second messenger

Shirin Fatma, Arpita Chakravarti, Xuankun Zeng, Raven H. Huang

This is a beautiful study that presents a first structure of CBASS effector Cap5 and provides an insight into regulation mechanism of this SAVED protein.

Cyclic-oligonucleotide-based anti-phage signaling systems (CBASS) are a family of defense systems against bacteriophages that share ancestry with the cGAS–STING innate immune pathway in animals. CBASS, like type III CRISPR-Cas systems, use small-molecule signaling as a way to transmit the information between the phage-sensing and cell-killing modules. CBASS systems are composed of an oligonucleotide cyclase, which generates signaling cyclic oligonucleotides (cOA) in response to phage infection, and an effector, that is activated by the cOA and promotes cell death. Effectors use SAVED sensor domain to detect signaling molecule and to activate their effector domain. SAVED domains are often found fused to various effector domains, including nucleases, proteases, phosphatases, NAD⁺ hydrolases and potentially pore-forming transmembrane proteins.

In this manuscript Fatma et al. present biochemical and structural characterization of two CBASS systems from *Asticcacaulis* sp. (As) and *Lactococcus lactis* (LI). Authors demonstrated that cyclase AsCdnG synthesizes 3'2'-cGAMP and this first time-observed signaling molecule activates both AsCap5 and LICap5 effectors for double-stranded DNA cleavage. So far, only the structure of Cap4 effector was available (Lowey et al. 2020). Authors succeed to solve the apo- structures of AsCap5 and LICap5 proteins. Both Cap5 effectors consist of N-terminal HNH nuclease domain and C-terminal SAVED domain. Interestingly, SAVED domain of Cap5 does not exactly match SAVED domain of Cap4. Determined structure of the LICap5-SAVED domain in the complex with 3'2'-cGAMP allowed understanding how the cGAMP molecule is recognized. Moreover, authors suggested a ligand-induced Back-to-Front SAVED-domain stacking mechanism for DNA recognition. Presumably it could be employed for activation of other CBASS effectors.

Comments:

1) HNH-type nucleases could use different metal ions (Mg^{2+} , Ni^{2+} , Co^{2+} , Mn^{2+} , Zn^{2+} , Ca^{2+}) as cofactors for double stranded DNA cleavage. Also some HNH nucleases retain some single-stranded DNA cleavage. How is about Cap5?

2) Authors could discuss DNA target specificity of Cap5.

Specific remarks:

1) Authors prove that CdnG cyclase intermediate product is ApG, not GpA using cleavage reaction with SVPD, where they observe cleavage products GMP and A (not G and AMP). This is provided in Extended Data Fig.2d which is cited late in the manuscript. Therefore for clarity please indicate in the legend of Figure 1(b,c) that "ApG is presumably intermediate product". Same, p. 10, Methods chapter, please change "ApG" to "intermediate ApG".

2) In Extended Data Fig.2d (above section) in P1 nuclease digestion experiment we can conclude only that reaction intermediate has 2',5' linkage, but we cannot discriminate between ApG/GpA in UPLC. A control UPLC chromatogram of reaction intermediate with "no nuclease added" should be included. Control UPLC chromatograms of synthetic A and GMP are required.

3) "Structures of effectors in complex with some of these signaling molecules are available. They include eukaryotic and bacterial STING10,24, bacterial endonuclease17, and CRISPR-associated RNA and DNA nucleases25–27." Please include citations of structures of CRISPR-associated effectors Card1 (Rostol et al 2021) and Csm6 (Jia et al 2019) in complex with cOAs.

4) cGAS-STING is a part of eukaryotic immune pathway. Please specify this in Introduction section.

Reviewer #2 (Remarks to the Author):

This manuscript by Fatma et. al. describes a detailed characterization of a new putative phage defense cyclic di-nucleotide signaling system in bacteria. The authors make several significant new findings. The authors describe the first mechanistic characterization of synthesis of 3',2'-cGAMP in bacteria. Also, although the Kranzchusch lab previously published the structure of a SAVED domain protein, this work is novel as it is the first structural characterization of a SAVED domain fused with the HNH-nuclease, which is the most common group of SAVED containing effectors. The authors propose a very interesting "front-to-back" model of activation whereby the 3',2'-cGAMP fuses opposite sides of SAVED domains. What is interesting about this model is the possibility that some SAVED effectors could form higher order oligomers that are fused with cyclic di-nucleotides. The data are clear, the experiments are well done, and the paper was very clearly written. The work

certainly adds new knowledge to the young field of CD-NTase signaling, and in my opinion will be of high impact. Thus, I only have a few minor comments.

1. Line 66-The authors state that the "capW" genes is in an operon with the CdnG/Cap5 systems that they study in this manuscript. Is there any data to conclude that the capW gene are actually part of an operon beyond their location in the genome? If not, then the authors can only speculate that they might be part of the cdnG/cap5 operon.
2. A new paper was published while this manuscript is in review that would be worth citing as it discovered 3',2'-cGAMP in Drosophila-PMID: 34261128
3. Fig 5f-the number designating lane 4 needs to be centered

Reviewer #3 (Remarks to the Author):

The manuscript "Molecular mechanisms of the CdnG-Cap5 antiphage defense system employing 3',2'-cGAMP as the second messenger" by Fatma... Huang presents a straightforward biochemical and structural analysis of a new subtype of CBASS system, an anti-bacteriophage defense pathway. In these pathways, phage infection stimulates synthesis of one of a variety of cyclic di- or trinucleotide signaling molecules by a class of cGAS-like enzymes, through an unknown mechanism. The signaling molecule in turn activates an effector protein, which usually kills the infected cell to abort the infection. A key insight of the past couple of years was the discovery that bacterial cGAS-like enzymes can synthesize a surprising variety of signaling molecules, using different RNA bases and with a mix of 2'-5' and 3'-5' linkages. In this work, the authors demonstrate that an as-yet uncharacterized family of bacterial cGAS-like enzymes generates a novel messenger molecule, and provide convincing evidence that this molecule activates the associated nuclease enzymes. The work is straightforward and overall well done. The study lacks a functional biological component to directly demonstrate the impact of their findings on viral defense, but given that these systems are fairly well-known by now, this should not detract too much from the significance. Overall, while the study is somewhat incremental, I think it's important to the field and I support publication overall.

Major points to address:

The two CBASS systems studied in this work each have three genes: CdnG (cGAS-like), Cap5 (SAVED-HNH nuclease), and a small protein the authors call "CapW", after a tryptophan-rich sequence that characterizes these proteins. The authors later state that their CBASS systems fall into the "Type II (short)" class of CBASS system as classified by Rotem Sorek and colleagues (reference 12). Thus it

seems likely that CapW is the short E2-like protein identified in these systems. If this is the case, the authors should clearly state that fact. It's unfortunate that Sorek and colleagues did not assign a "CapX" name to this short E2-like gene in their work; in the absence of that, I suppose CapW is an OK name. I would suggest, however, that the authors assign a name with "Cap" and then a number, as seems to be the developing consensus for gene names in CBASS systems. Or alternatively (and I favor this strategy), avoid naming the protein at all since the authors don't characterize it in this work. Simply call it "E2-like" and don't mention after that.

The authors spend some effort modeling a Cap5-substrate complex based on a structure of a Cas9 HNH domain bound to an RNA-DNA hybrid. But, this is not the correct substrate for Cap5, which as the authors nicely demonstrate, cleaves dsDNA. The modeling therefore is not very useful, especially as the modeled RNA-DNA hybrid is an A-form helix while dsDNA is B-form. The authors can't really conclude anything about the likely activity level of their visualized conformation of Cap5 based on this modeling. I would therefore change the sentences in lines 174-177.

In the authors' structure of the SAVED domain bound to cGAMP, is there by chance a crystal packing interaction that might demonstrate how the SAVED domain dimerizes in response to binding?

Minor points:

Line 173 - "AsCas9-HNH" should read "AsCap5-HNH"

Line 236 - typo - K346A should be K347A

Hard to draw any conclusions from modeling Cap5 bound to an RNA-DNA hybrid - this is not the relevant substrate and does not actually even fit on the structure...

Lines 174-177 - can't conclude that this conformation of Cap5 is "inactive" based on modeling with the wrong substrate.

Figure 2 - please label the DNA ladder in at least one panel.

RESPONSE TO REVIEWER COMMENTS

Reviewer #1 (Remarks to the Author):

Molecular mechanisms of the CdnG-Cap5 antiphage defense system employing 2 3',2'-cGAMP as the second messenger

Shirin Fatma, Arpita Chakravarti, Xuankun Zeng, Raven H. Huang

This is a beautiful study that presents a first structure of CBASS effector Cap5 and provides an insight into regulation mechanism of this SAVED protein.

Cyclic-oligonucleotide-based anti-phage signaling systems (CBASS) are a family of defense systems against bacteriophages that share ancestry with the cGAS–STING innate immune pathway in animals. CBASS, like type III CRISPR-Cas systems, use small-molecule signaling as a way to transmit the information between the phage-sensing and cell-killing modules. CBASS systems are composed of an oligonucleotide cyclase, which generates signaling cyclic oligonucleotides (cOA) in response to phage infection, and an effector, that is activated by the cOA and promotes cell death. Effectors use SAVED sensor domain to detect signaling molecule and to activate their effector domain. SAVED domains are often found fused to various effector domains, including nucleases, proteases, phosphatases, NAD⁺ hydrolases and potentially pore-forming transmembrane proteins.

In this manuscript Fatma et al. present biochemical and structural characterization of two CBASS systems from *Asticcacaulis* sp. (As) and *Lactococcus lactis* (LI). Authors demonstrated that cyclase AsCdnG synthesizes 3'2'-cGAMP and this first time-observed signaling molecule activates both AsCap5 and LICap5 effectors for double-stranded DNA cleavage. So far, only the structure of Cap4 effector was available (Lowey et al. 2020). Authors succeed to solve the apo- structures of AsCap5 and LICap5 proteins. Both Cap5 effectors consist of N-terminal HNH nuclease domain and C-terminal SAVED domain. Interestingly, SAVED domain of Cap5 does not exactly match SAVED domain of Cap4. Determined structure of the LICap5-SAVED domain in the complex with 3'2'-cGAMP allowed understanding how the cGAMP molecule is recognized. Moreover, authors suggested a ligand-induced Back-to-Front SAVED-domain stacking mechanism for DNA recognition. Presumably it could be employed for activation of other CBASS effectors.

Comments:

1) HNH-type nucleases could use different metal ions (Mg²⁺, Ni²⁺, Co²⁺, Mn²⁺, Zn²⁺, Ca²⁺) as cofactors for double stranded DNA cleavage. Also some HNH nucleases retain some single-stranded DNA cleavage. How is about Cap5?

To prepare for the revision, we have spent most of our effort to carry out experiments to address divalent ion specificity of Cap5. Unfortunately, the experiments were not successful. The likely cause is the fact that Cap5 requires not only a divalent ion in the active site, but also a Zn²⁺ ion for the zinc finger

that plays an important structural role. We first removed all divalent ions from Cap5 with EDTA. But we were not able to reconstitute the enzymatic activity of Cap5 with the addition of Zn²⁺ and various other divalent ions (Mg²⁺, Mn²⁺, Ni²⁺, Co²⁺, and Ca²⁺). We speculate that the removal of Zn²⁺ ion causes structural damage of Cap5, which could not be restored by the addition of the external Zn²⁺ ion. We then carried out gel filtration of Cap5 in a buffer that lacks the divalent ions, hoping to remove the divalent ion (most likely Mg²⁺) in the active site but to maintain Zn²⁺ ion. However, Cap5 obtained from gel filtration was as active as the one prior chromatography, indicating that the divalent ion in the active site was retained.

As whether Cap5 retains the ability to degrade single-stranded DNA, we decided not to pursue further for two reasons. First, it is difficult to obtain a long single-stranded DNA as the potential substrate. Second, based on our study presented in the manuscript, it is very unlikely that single-stranded DNAs are biological substrates of Cap5.

2) Authors could discuss DNA target specificity of Cap5.

To obtain information suggested by the reviewer, extensive experiments, including isolation of cleaved products, high throughput DNA sequencing, and extensive data analysis of the resulting sequencing data, are required. Because this additional information does not contribute significantly to the biological function of Cap5 (efficient DNA degradation irrespective of DNA sequences), we decided not to pursue it.

Specific remarks:

1) Authors prove that CdnG cyclase intermediate product is ApG, not GpA using cleavage reaction with SVPD, where they observe cleavage products GMP and A (not G and AMP). This is provided in Extended Data Fig.2d which is cited late in the manuscript. Therefore for clarity please indicate in the legend of Figure 1(b,c) that “ApG is presumably intermediate product”. Same, p. 10, Methods chapter, please change “ApG” to “intermediate ApG”.

We have made the changes suggested by the reviewer.

2) In Extended Data Fig.2d (above section) in P1 nuclease digestion experiment we can conclude only that reaction intermediate has 2',5' linkage, but we cannot discriminate between ApG/GpA in UPLC. A control UPLC chromatogram of reaction intermediate with "no nuclease added" should be included. Control UPLC chromatograms of synthetic A and GMP are required.

We have carried out additional experiments and added these two controls in Fig. 2d.

3) “Structures of effectors in complex with some of these signaling molecules are available. They include eukaryotic and bacterial STING^{10,24}, bacterial endonuclease¹⁷, and CRISPR-associated RNA and DNA nucleases^{25–27}.” Please include citations of structures of CRISPR-associated effectors Card1 (Rostol et al 2021) and Csm6 (Jia et al 2019) in complex with cOAs.

We have added these two citations.

4) cGAS-STING is a part of eukaryotic immune pathway. Please specify this in Introduction section.

We would like further clarification from the reviewer, as we have dedicated the first paragraph of Introduction section to describe cGAS and STING.

Reviewer #2 (Remarks to the Author):

This manuscript by Fatma et. al. describes a detailed characterization of a new putative phage defense cyclic di-nucleotide signaling system in bacteria. The authors make several significant new findings. The authors describe the first mechanistic characterization of synthesis of 3',2'-cGAMP in bacteria. Also, although the Kranzusch lab previously published the structure of a SAVED domain protein, this work is novel as it is the first structural characterization of a SAVED domain fused with the HNH-nuclease, which is the most common group of SAVED containing effectors. The authors propose a very interesting “front-to-back” model of activation whereby the 3',2'-cGAMP fuses opposite sides of SAVED domains. What is interesting about this model is the possibility that some SAVED effectors could form higher order oligomers that are fused with cyclic di-nucleotides. The data are clear, the experiments are well done, and the paper was very clearly written. The work certainly adds new knowledge to the young field of CD-NTase signaling, and in my opinion will be of high impact. Thus, I only have a few minor comments.

1. Line 66-The authors state that the “capW” genes is in an operon with the CdnG/Cap5 systems that they study in this manuscript. Is there any data to conclude that the capW gene are actually part of an operon beyond their location in the genome? If not, then the authors can only speculate that they might be part of the cdnG/cap5 operon.

Based on our bioinformatic analysis, there are ~500 CapW unique sequences, and most of their encoding genes are the immediate neighbors of the genes encoding NTases. Therefore, capW gene is very likely to be part of the operon. Nevertheless, we have added “might” in the revision.

2. A new paper was published while this manuscript is in review that would be worth citing as it discovered 3',2'-cGAMP in Drosophila-PMID: 34261128

We have cited both *Nature* papers describing the discovery of 3',2'-cGAMP in Drosophila in Discussion section.

3. Fig 5f-the number designating lane 4 needs to be centered

We have corrected the error.

Reviewer #3 (Remarks to the Author):

The manuscript "Molecular mechanisms of the CdnG-Cap5 antiphage defense system employing 3',2'-cGAMP as the second messenger" by Fatma... Huang presents a straightforward biochemical and structural analysis of a new subtype of CBASS system, an anti-bacteriophage defense pathway. In these pathways, phage infection stimulates synthesis of one of a variety of cyclic di- or trinucleotide signaling molecules by a class of cGAS-like enzymes, through an unknown mechanism. The signaling molecule in turn activates an effector protein, which usually kills the infected cell to abort the infection. A key insight of the past couple of years was the discovery that bacterial cGAS-like enzymes can synthesize a surprising variety of signaling molecules, using different RNA bases and with a mix of 2'-5' and 3'-5' linkages. In this work, the authors demonstrate that an as-yet uncharacterized family of bacterial cGAS-like enzymes generates a novel messenger molecule, and provide convincing evidence that this molecule activates the associated nuclease enzymes. The work is straightforward and overall well done. The study lacks a functional biological component to directly demonstrate the impact of their findings on viral defense, but given that these systems are fairly well-known by now, this should not detract too much from the significance. Overall, while the study is somewhat incremental, I think it's important to the field and I support publication overall.

Major points to address:

The two CBASS systems studied in this work each have three genes: CdnG (cGAS-like), Cap5 (SAVED-HNH nuclease), and a small protein the authors call "CapW", after a tryptophan-rich sequence that characterizes these proteins. The authors later state that their CBASS systems fall into the "Type II (short)" class of CBASS system as classified by Rotem Sorek and colleagues (reference 12). Thus it seems likely that CapW is the short E2-like protein identified in these systems. If this is the case, the authors should clearly state that fact. It's unfortunate that Sorek and colleagues did not assign a "CapX" name to this short E2-like gene in their work; in the absence of that, I suppose CapW is an OK name. I would suggest, however, that the authors assign a name with "Cap" and then a number, as seems to be the developing consensus for gene names in CBASS systems. Or alternatively (and I favor this strategy), avoid naming the protein at all since the authors don't characterize it in this work. Simply call it "E2-like" and don't mention after that.

CapW belongs to a completely different family of proteins without a Pfam number. Also see our response to reviewer 2 regarding CapW. During our study, we also observed that some CdnG-Cap5 systems are encoded in three-gene operons with the third gene encoding short-E2-like proteins, which belong to PF14457. Therefore, based on sequence comparison as well as whether they have

Pfam numbers, CapW and short-E2-like are completely different proteins. To make it clear, we have added a long sentence to explain their difference in Introduction section.

We still prefer our TENTATIVE assignment of CapW, as it gives this family of proteins some character. In an event that CapW is further characterized experimentally, W can be readily converted to a number to give it a formal name.

The authors spend some effort modeling a Cap5-substrate complex based on a structure of a Cas9 HNH domain bound to an RNA-DNA hybrid. But, this is not the correct substrate for Cap5, which as the authors nicely demonstrate, cleaves dsDNA. The modeling therefore is not very useful, especially as the modeled RNA-DNA hybrid is an A-form helix while dsDNA is B-form. The authors can't really conclude anything about the likely activity level of their visualized conformation of Cap5 based on this modeling. I would therefore change the sentences in lines 174-177.

We modified the sentences in lines 174-177 to acknowledge the difference of the modeled DNA•RNA hybrid and the actual substrate of Cap5. But we slightly disagree with the reviewer as we believe the comparison of Cap5 to Cas9, including the modeled RNA•DNA hybrid from Cas9, is still useful. The structural difference of A- and B-form DNAs is small, in particular at the local backbone conformation near the active sites of nucleases.

In the authors' structure of the SAVED domain bound to cGAMP, is there by chance a crystal packing interaction that might demonstrate how the SAVED domain dimerizes in response to binding?

The crystal packing reveals that cGAMP faces a wide-open space without other SAVED domains nearby. Therefore, the packing does not provide insight into dimerization.

Minor points:

Line 173 - "AsCas9-HNH" should read "AsCap5-HNH"

Corrected.

Line 236 - typo - K346A should be K347A

Corrected.

Hard to draw any conclusions from modeling Cap5 bound to an RNA-DNA hybrid - this is not the relevant substrate and does not actually even fit on the structure...

Lines 174-177 - can't conclude that this conformation of Cap5 is "inactive" based on modeling with the wrong substrate.

As described earlier, we have modified the description. We hope our new description is acceptable to the reviewer.

Figure 2 - please label the DNA ladder in at least one panel.

We have labeled DNA ladder in all panels of gels.